# A Case Series of Gastric Metastatic Growths

**DOI:** 10.3390/diseases10030061

**Published:** 2022-09-07

**Authors:** Mustafa Gandhi, Harleen Kaur Chela, Hamza Ertugrul, Alhareth Al Juboori, Karthik Gangu, Deepthi Rao, Ebubekir Daglilar

**Affiliations:** 1Department of Medicine, University of Missouri, Columbia, MO 65201, USA; 2Department of Gastroenterology, University of Missouri, Columbia, MO 65201, USA; 3Department of Medicine, Division of Hospital Medicine, University of Kansas Medical Center, Kansas City, KS 66160, USA; 4Department of Pathology, University of Missouri, Columbia, MO 65201, USA; 5Department of Gastroenterology, University of West Virginia, Charleston, WV 25304, USA

**Keywords:** gastroenterology, endoscopy, cancer, metastasis, stomach

## Abstract

Gastric cancer is one of the gastrointestinal malignancies that can be quite devastating with high morbidity and mortality. Unfortunately, it is a malignancy that is encountered all across the world and is often brought into suspicion based on symptoms of the patient. The presentation differs based on the symptomatology and can be quite variable in each and every case. Malignant lesions in the stomach discovered endoscopically can represent as primary gastric growths or can be secondary as a consequence of metastatic spread from a distant primary site. It is important to recognize the different patterns of presentation of metastatic disease and to be aware of the primary tumor sites. The treatment and ultimately the prognosis changes drastically when dealing with a metastatic disease as opposed to a primary localized source with limited spread. The aim of our study is to present a mini series of cases that manifest as metastatic gastric growths. Their clinical, endoscopic and histological appearance is depicted to provide an understanding of each case. The primary sites of origin for our patients were the lungs, skin, lymphoid tissue and kidneys. Their overall clinical course is presented including the approach to the management in each case as well as their outcomes.

## 1. Introduction

With an incidence rate of 0.2–0.7% based on clinical and autopsy reports, gastric metastases are rare whereas primary gastric malignancy is one of the most common cancers worldwide and a leading cause of cancer-related death [1,2]. The most common primary sites of metastases to the stomach as described in the literature are the breast, melanoma, lung, kidney, and esophagus [1,3]. It can be difficult to differentiate primary gastric malignancy from metastatic gastric lesions based on clinical, radiological, endoscopic, and histopathological features. However, it is important as to recognize this as metastatic tumors in the stomach are being encountered more frequently [1,4].

For gastric metastatic lesions, although the prognosis may be guarded and the incidence rare, it is valuable to recognize the variety of possible presentations for prompt and accurate diagnosis and treatment. Not only is the distinction among various metastatic lesions beneficial, but also the differentiation of a metastatic gastric malignancy from primary gastric cancer. The aim of our study is to show the various clinicopathological presentations of metastatic lesions in the stomach along with their outcomes.

## 2. Methods

We analyzed six patients with metastatic cancer in the stomach that presented to the University of Missouri Hospital, Columbia between 2019 and 2021 and detected in the Department of Gastroenterology and Hepatology. These patients underwent esophagogastroduodenoscopy (EGD) either for evaluation of gastrointestinal symptoms or for diagnosis of a gastric lesion detected on imaging. Diagnosis of metastatic gastric cancer was made by computed tomography (CT), magnetic resonance imaging (MRI), ultrasonography of the abdomen, EGD, analysis of biopsy specimens, and positron emission tomography. We further reviewed the clinical course and therapy for each of these patients. The reports include solid organ malignancy and lymphomas.

## 3. Results

### 3.1. Patient Characteristics

Six patients who had been diagnosed with metastatic gastric tumors arising from other malignancies were included in the present study. The clinical features of these six patients are listed in Table 1. Three patients (50%) were men and three (50%) were women, with patient age ranging from 35 to 84 years (median, 64). Four patients had lesions in the fundus; with three out of these having additional lesions in the body, cardia or both. One patient had a lesion in the body of the stomach, while the location had not been specified for one of the patients. The lesions had varied appearances from ulcers to large fungating masses. The common additional sites of metastasis included the liver and esophagus while the lungs, urinary bladder, bones, and mediastinum were less frequent locations. Two cases had primary cancer originating in the lungs, two patients had lymphoma while the remaining cases were primary renal cancer and melanoma. The metastases presented as multiple lesions in 3 cases (50%) while they presented as solitary lesions in the other 3 cases (50%). For the treatment of the malignancy, one patient was treated with chemotherapy and immunotherapy, while three received chemotherapy with additional immunotherapy in two of those. One of the patients also had surgery and two of the patients opted to forego treatment due to the advanced nature of the disease in favor of palliative care. The outcomes for the patients were poor, with five of them passing within 5 days to 2 months after the EGD diagnosis of gastric metastases. One patient had disease progression based on the latest scans.

A 35-year-old with chronic tobacco use presented initially with hemoptysis and dyspnea and was found to have a large infiltrative soft tissue mass arising from the mediastinum, extending into the right hilum and right middle lobe. Bronchoscopy was performed with the aspiration of mediastinal lymph nodes revealing poorly differentiated carcinoma. Due to diffuse gastric wall thickening with markedly increased FDG uptake on PET CT (Figure 1), an upper endoscopy was performed which showed a large fungating gastric fundal mass (Figure 2). Multiple single pass biopsies were obtained from this mass. Biopsies revealed poorly differentiated carcinoma and staining was negative for Helicobacter pylori (Figure 3). Eventually, a biopsy was performed of right-sided supraclavicular lymph node that showed metastasis of poorly differentiated large cell carcinoma with focal chromogranin and strong TTF-1 expression.

A 73-year-old female was initially diagnosed with left-sided renal cell carcinoma and underwent nephrectomy, however few years later developed a right-sided renal mass along with pulmonary nodules. She underwent partial right nephrectomy with right upper lobe VATS (showed primary as clear cell RCC). Was started on sunitinib, then switched to pazopanib due to severe mucositis. Eventually several years later developed metastatic liver lesions requiring microwave ablation and subsequently she presented with melena. EGD showed several anterior and posterior wall gastric polyps, some of which were noted to be actively oozing. They were removed with hot snare technique. The pathology was reported to be metastatic renal cell carcinoma clear cell type in a background of the hyperplastic gastric mucosa (Figure 4 and Figure 5). She was on axitinib and required another endoscopy later on for hematemesis and several large gastric masses were seen that were friable and oozed readily on contact with endoscope (Figure 6).

A 61-year-old male with recurrent marginal zone lymphoma on chemotherapy presented for endoscopic evaluation for iron deficiency anemia. Colonoscopy revealed tubular adenomas, upper endoscopy revealed a clean based gastric ulcer and gastric erosions (Figure 7). Sloughing was also noted in mid esophagus with biopsies consistent with esophagitis dissecans superficialis (Figure 8). The gastric ulcer biopsies showed ulceration with granulation tissue along with large, atypical cells concerning for lymphoma (Figure 9 and Figure 10).

An 84-year-old male with multiple comorbidities presented with early satiety and weakness presented for evaluation for a 6 cm gastric mass noted on a CT scan. EGD revealed about 5 cm large ulcerated, necrotic gastric mass with extension to about 1 cm below the GE junction (Figure 11). Additional gastric nodules were seen concerning for satellite lesions. Biopsies from the gastric mass were consistent with diffuse large B-cell lymphoma (Figure 12 and Figure 13). 

A 66-year-old female presented for endoscopic evaluation of progressive dysphagia to solids and liquids. CT scan showed a heterogeneous necrotic mass in the upper mediastinum with sub-carinal and left hilar masses along with mass-effect and encasement of mediastinal structures as well. EGD showed a large eroding posterior mediastinal mass was seen starting 22 cm from incisors (Figure 14). Large amount of fluid/pus in the cavity which suctioned easily. The stomach had innumerable polypoid-appearing lesions with superficial ulcers/erosions (Figure 15). Antrum was relatively spared. Nodular duodenitis was seen in the examined portion of the duodenum. Gastric biopsies revealed changes consistent with malignant melanoma (Figure 16).

A 76-year-old with multiple medical comorbidities presented with melena for which endosocpic evaluation was requested. Imaging had revealed concerns for diffuse hepatic metastases in addition to bone metastases with large right perihilar mass with bilateral pulmonary nodules, mediastinal adenopathy and right sided pleural effusion. EGD showed several cratered gastric body ulcers a few of them actively oozing (Figure 17). Hemostasis achieved with bipolar coagulation and biopsies obtained from non-bleeding ulcers. Pathology revealed metastatic poorly differentiated neuroendocrine carcinoma consistent with small cell carcinoma (Figure 18). 

### 3.2. Treatment of Gastric Metastases

One patient with malignant melanoma was initially treated with radiotherapy and immunotherapy, but after the EGD diagnosis of metastases in the stomach with additional spread to the esophagus, pancreas, duodenum, and mediastinum, they opted for palliative care in view of the advanced disease and then succumbed 5 days later. The patient with diffuse large B-cell lymphoma which had also spread to the liver and urinary bladder also chose the palliative route of treatment and passed away 2 months after the EGD diagnosis of gastric metastasis. Chemotherapy with carboplatin and etoposide was used in the treatment of metastatic large cell neuroendocrine carcinoma of the lung but despite therapy, the patient died one month after diagnosis of metastasis. Another case with metastatic small cell carcinoma of the lung received only palliative care and passed within 7 days after EGD. In the case of the metastatic renal cell carcinoma, the patient had a prior nephrectomy and subsequent treatment with chemotherapy and immunotherapy (Axitinib). Despite this therapy, he had metastasis to the stomach, and liver lungs and died 1 week late the EGD. Marginal zone lymphoma in the stomach was treated with Rituximab, Zanubrutinib, and Umbrasilib. Treatment failure with those therapies prompted the O-CHOP regimen (Cyclophosphamide-Doxorubicin-Obinutuzumab-Prednisolone-Vincristine) most recently with the latest imaging studies indicative of disease progression. 

## 4. Discussion

As described in the literature, some of the common malignancies that metastasize to the stomach are the breast, melanoma, lung, esophagus, and kidney which concurs to an extent with the cases described in our study [5]. Even though the exact mechanism of gastric metastasis has not been described some of the probable pathways include direct tumor invasion, lymphatic spread, hematogenous dissemination, and peritoneal spread. Specifically, during hematogenous dissemination, tumor cells likely become trapped within the submucosa or the serosa which are known to have a rich blood supply [3].

Melanoma is among the most common malignancies to metastasize to the gastrointestinal (GI) tract in general. The primary tumor is usually a cutaneous source and less frequently ocular in origin. On rare occasions, melanomas in the GI tract, including the stomach can be primary tumors [6]. Although the tumor appearance on endoscopy is roughly classified into three categories: submucosal masses with ulcerations, mass lesions with necrosis and melanosis, and ulcerated melanotic nodules arising on normal rugae; the classic appearance is described as multiple, small, pigmented nodules that may ulcerate to produce a ‘target-like or ‘bull’s eye’ lesion [7]. In concurrence with the case described in our study, most melanoma metastases occur in the body and fundus of the stomach [8]. In a retrospective review of 124 patients who had melanoma with GI metastases, the median survival for those who underwent resection of the GI tract was 48.9 months compared with 5.7 months in those who underwent palliative and non-surgical interventions [9]. In our study, the patient presented with an endoscopic appearance of polypoid lesions with ulcerations and simultaneous spread to multiple other organs, lending a poor prognosis.

Some of the frequently encountered sites of extra-nodal NHL include the central nervous system, gastric, intestinal, and skin. The stomach is the most common site of extra-nodal non-Hodgkin’s lymphoma (NHL) representing between 30% and 40% of all extra-nodal lymphomas and 55% and 65% of all gastrointestinal lymphomas [10]. In our study, we describe two cases of lymphoma metastasis to the stomach with the diffuse large B-cell lymphoma (DLBCL) presenting as an ulcerated and necrotic mass while the marginal zone lymphoma (MZL) appeared as a clean-based ulcer on endoscopy. Although the literature is available for primary gastric/gastrointestinal lymphomas, there is a lack of abundant literature for lymphomas metastasizing to the stomach or GI tract.

Lung cancer metastasis to the gastrointestinal tract has an incidence of 0.5–10% and most commonly occurs in the small bowel with gastric metastases being rare with an incidence to range between 0.2 and 1.7% as reported by autopsy data [11]. Compared with non-small cell lung cancer (NSCLC), small cell lung cancer more commonly results in metastasis to the stomach. Gastrointestinal metastasis can rarely lead to complications such as obstruction, bleeding, or perforation [12]. Gastric metastasis in lung cancer is hypothesized to occur via swallowing of sputum containing abundant cancer cells, which then travel into the digestive tract; this mechanism is especially relevant in smokers who are more susceptible to gastric mucosal damage compared to nonsmokers [13]. However, the validity of this hypothesis still needs to be confirmed. Both the patients in our study, had a very poor prognosis, with death occurring at 7 days for squamous cell carcinoma and at 1 month for the large cell cancer. The reported prognosis of gastric metastasis in the literature for lung cancer has been very poor. Kim et al. reported that the median survival time in pulmonary carcinoma with gastrointestinal metastasis was 94.5 days, ranging from 12 to 1907 days [14].

A limitation of our present study is the relatively small number of patients from a single institution. Further studies with a larger sample size are necessary to obtain a better understanding of the varied presentations of gastric metastases due to the malignancy of different organs.

## 5. Conclusions

To conclude, physicians should be cognizant of the possible existence of metastatic gastric cancer and be able to differentiate from other primary malignancies. This is especially important for prompt evaluation, diagnosis, and therapy as the pathways for treatment are quite diverse depending on whether the malignancy is primary or one that has originated from a distant site. Distinguishing a metastatic process from one that is primary is crucial as the management would not only vary drastically along with the prognosis itself. As metastasis of any sort itself carries a grim prognosis as compared to a primary disease process that is limited to the organ of origin. Appropriate systemic treatment of the primary cancer with chemotherapy, immunotherapy, or hormonal treatment is the preferred treatment for gastric metastasis. Surgical resection of metastatic gastric tumors may be beneficial to improve patients’ quality of life when there is a solitary lesion or a high risk of complications like bleeding and perforation.

## Figures and Tables

**Figure 1 diseases-10-00061-f001:**
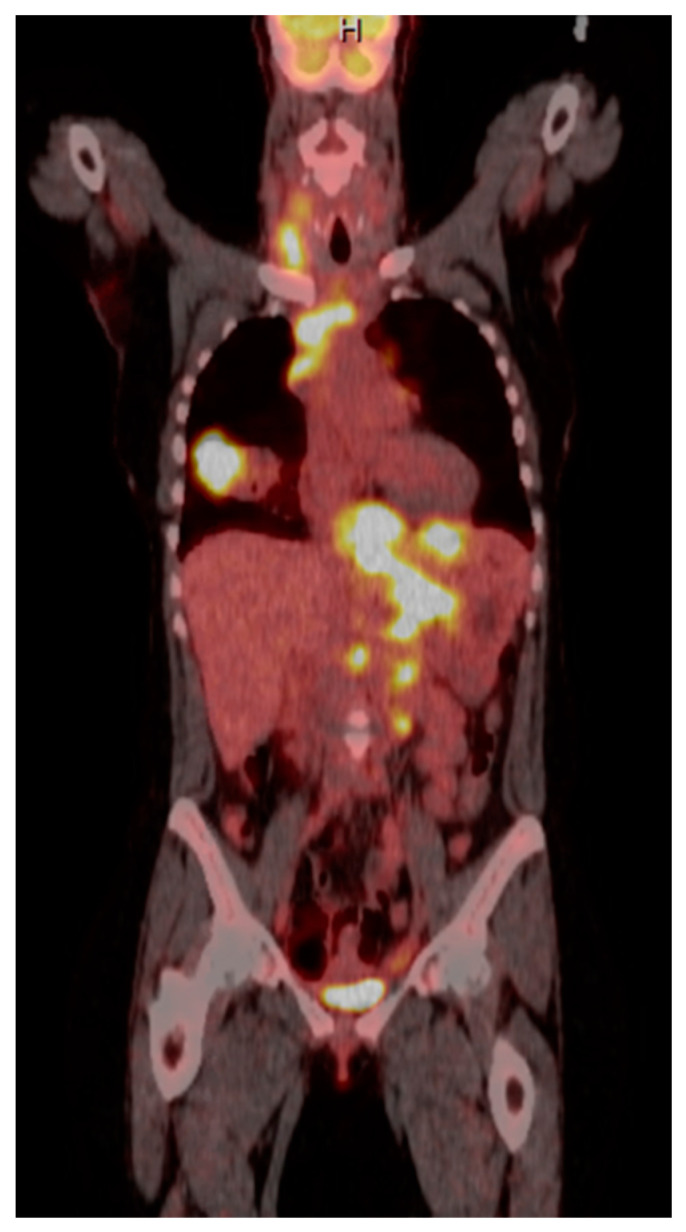
PET CT scan showing diffuse gastric wall thickening.

**Figure 2 diseases-10-00061-f002:**
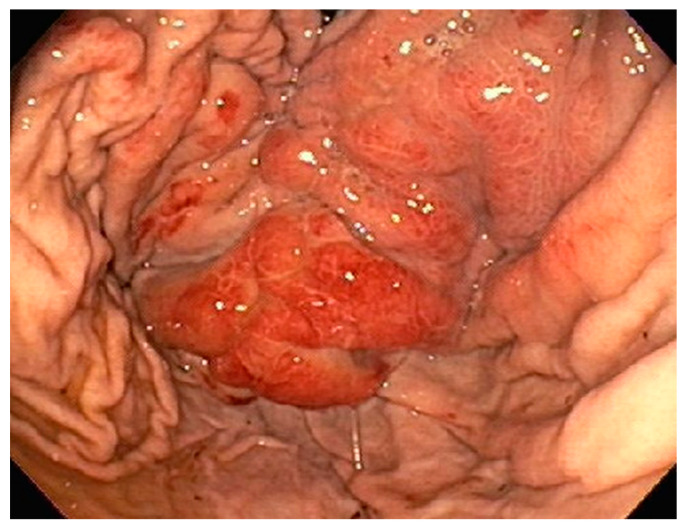
EGD revealing a large fungating gastric fundal mass.

**Figure 3 diseases-10-00061-f003:**
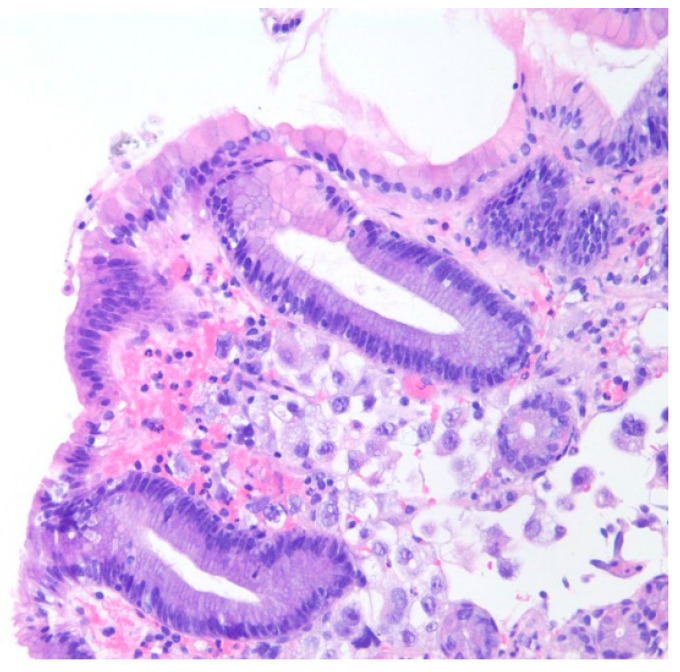
Poorly differentiated carcinoma cells are seen at the mucosal neck region of the gastric foveolar epithelium on hematoxylin and eosin stain at 200× magnification.

**Figure 4 diseases-10-00061-f004:**
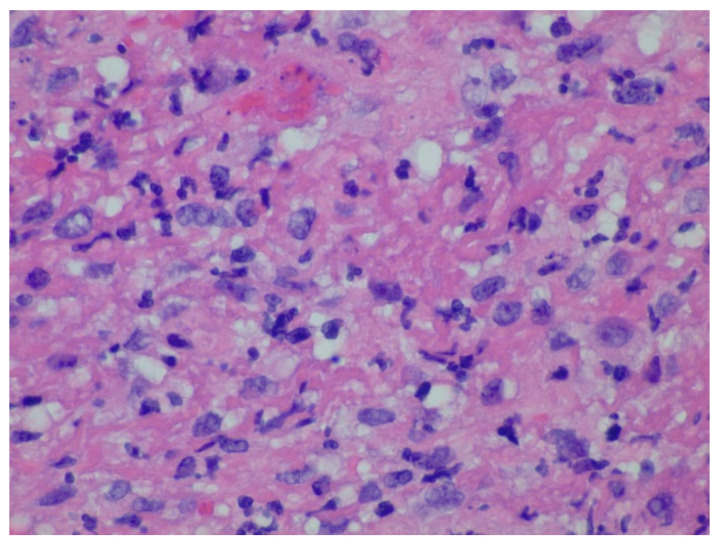
Diffuse infiltration of Renal cell carcinoma cells, clear cell type, with clear and granular eosinophilic cytoplasm in a background of gastric lamina propria seen on Hematoxylin and eosin stain at 600× magnification.

**Figure 5 diseases-10-00061-f005:**
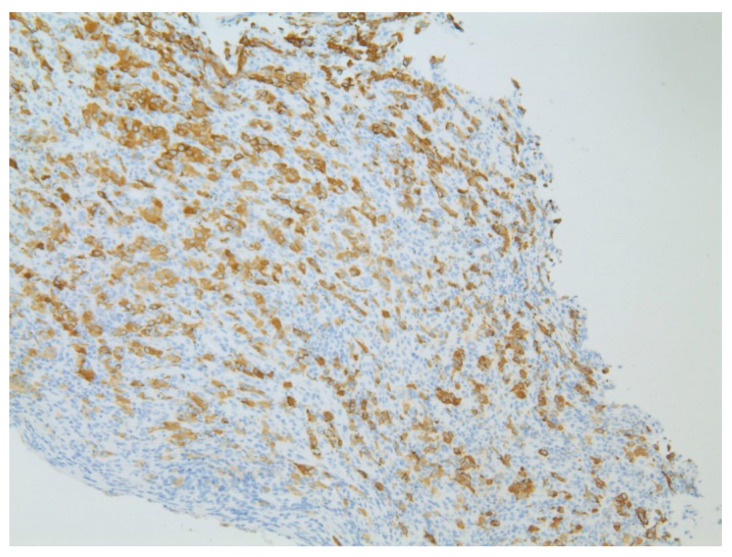
Cytokeratin AE1/AE3 immunostained section showing diffuse infiltration of metastatic renal carcinoma cells (positive expression) noted in a background of gastric stroma (negative expression) at 100× magnification.

**Figure 6 diseases-10-00061-f006:**
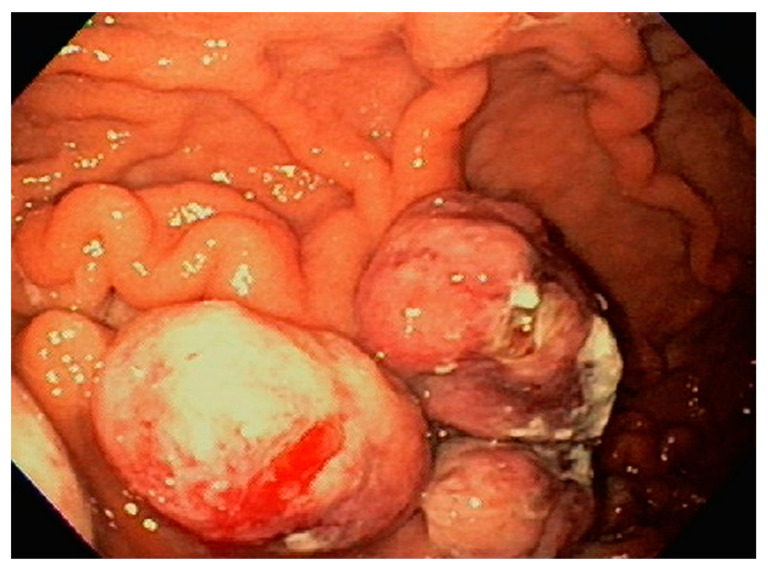
Large friable gastric masses seen on endoscopy.

**Figure 7 diseases-10-00061-f007:**
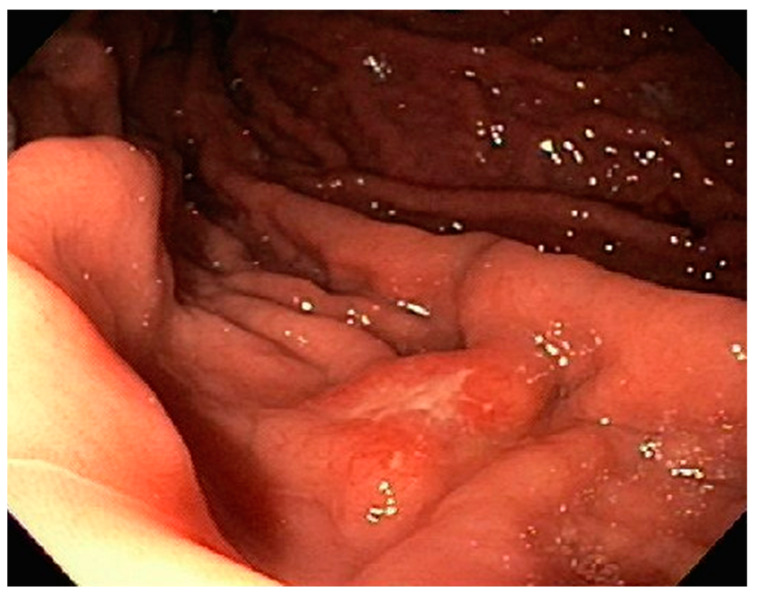
Clean based gastric ulcer noted on endoscopy.

**Figure 8 diseases-10-00061-f008:**
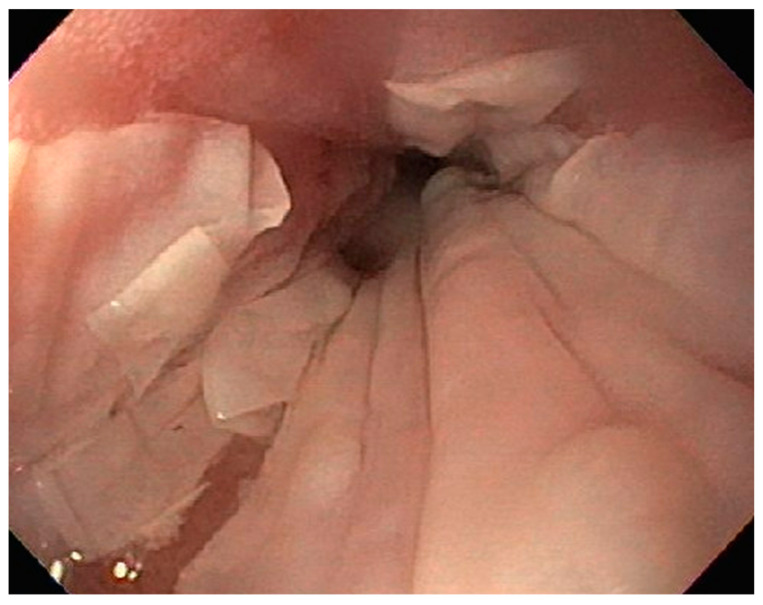
Sloughing in mid esophagus noted on endoscopy.

**Figure 9 diseases-10-00061-f009:**
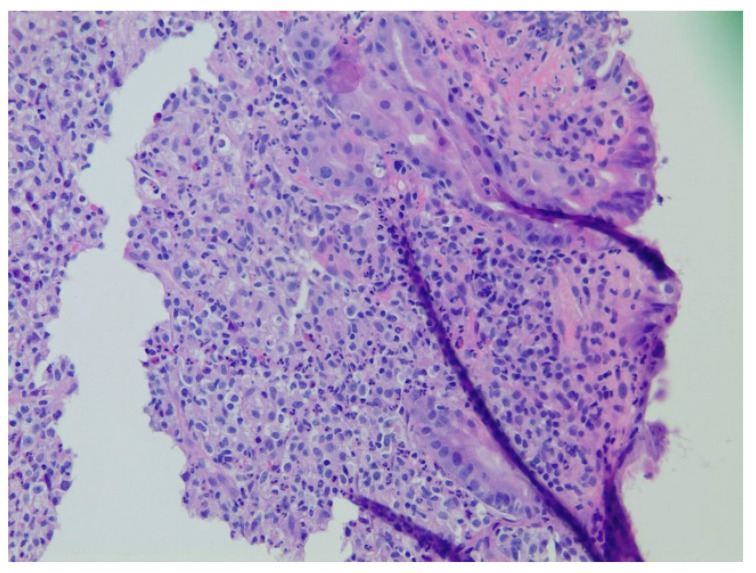
Predominantly small to medium-sized lymphocytes with pale cytoplasm (centrocyte-like) with slightly irregular nuclear contours with inconspicuous nucleoli and relatively abundant cytoplasm concerning for lymphoma noted on hematoxylin and eosin stain at 200× magnification.

**Figure 10 diseases-10-00061-f010:**
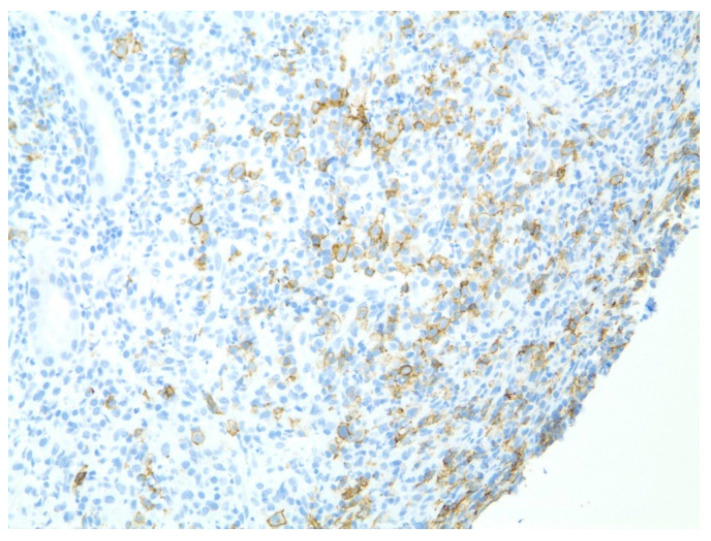
Immunohistochemical stain CD20 stained sections showing atypical large B-cells suggestive of marginal zone lymphoma with positive CD20 expression (seen at 200× magnification).

**Figure 11 diseases-10-00061-f011:**
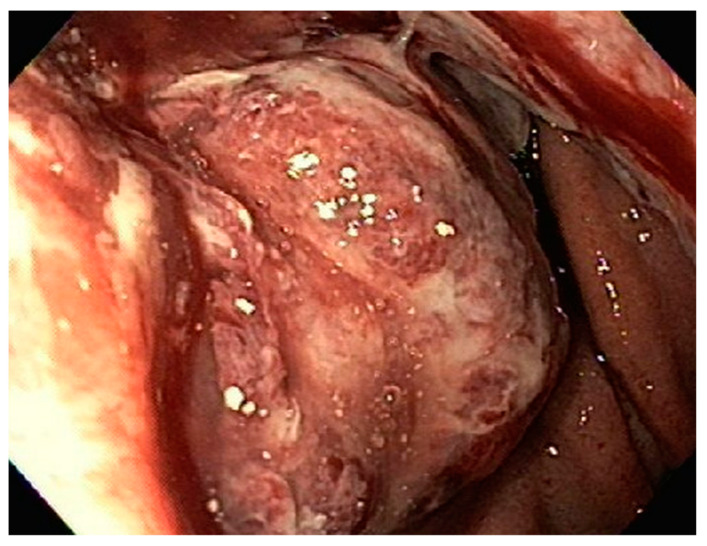
Large necrotic, ulcerated gastric mass noted on upper endoscopy.

**Figure 12 diseases-10-00061-f012:**
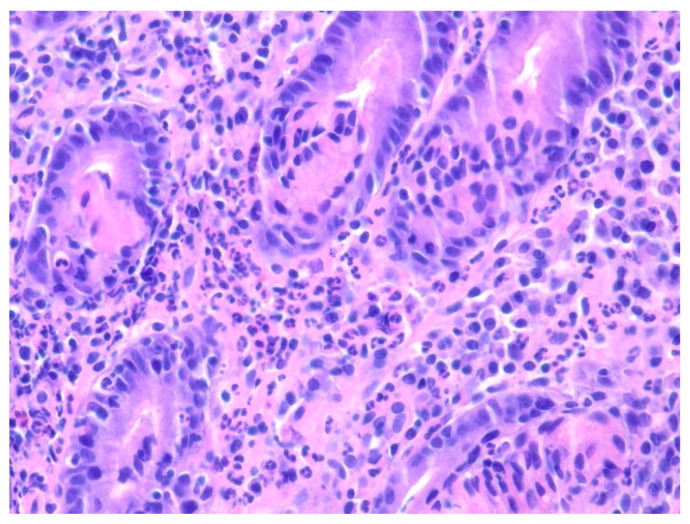
Diffuse proliferation of large pleomorphic lymphoid cells with prominent nucleoli with some nuclei being lobulatedThere is a background of small lymphocytes, plasma cells, and neutrophils within the gastric mucosa on Hematoxylin and eosin stain at 400× magnification.

**Figure 13 diseases-10-00061-f013:**
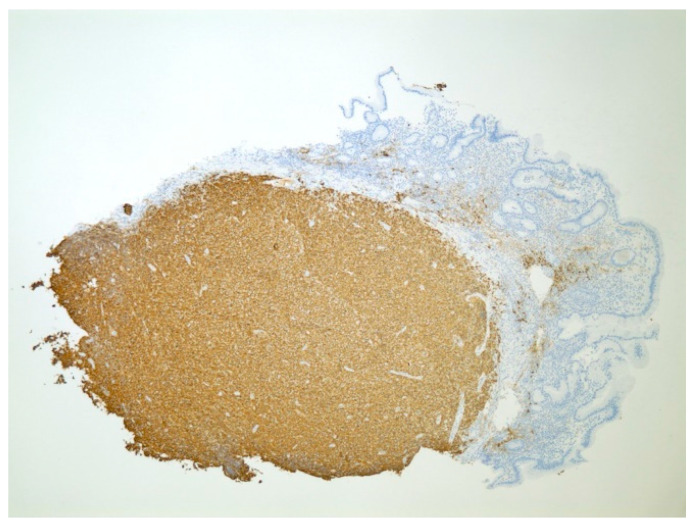
Immunohistochemical stain for CD20 showing diffuse positivity at 40× magnification.

**Figure 14 diseases-10-00061-f014:**
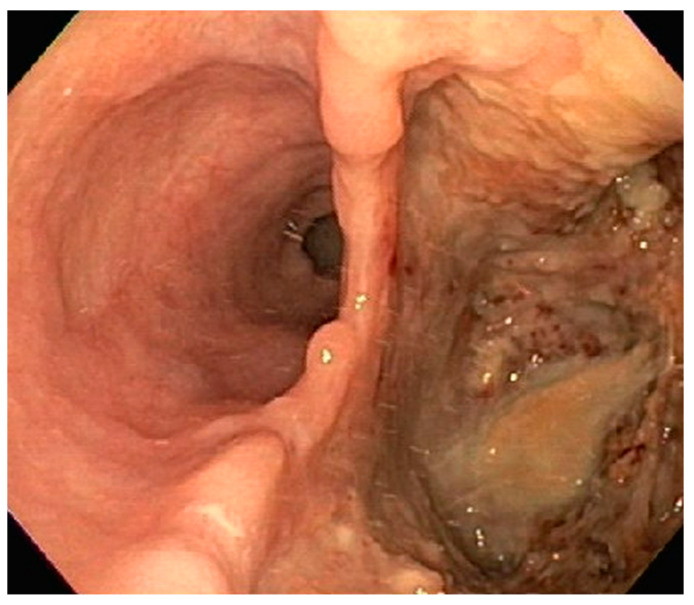
Large erosive mass noted on EGD about 22 cm from incisors.

**Figure 15 diseases-10-00061-f015:**
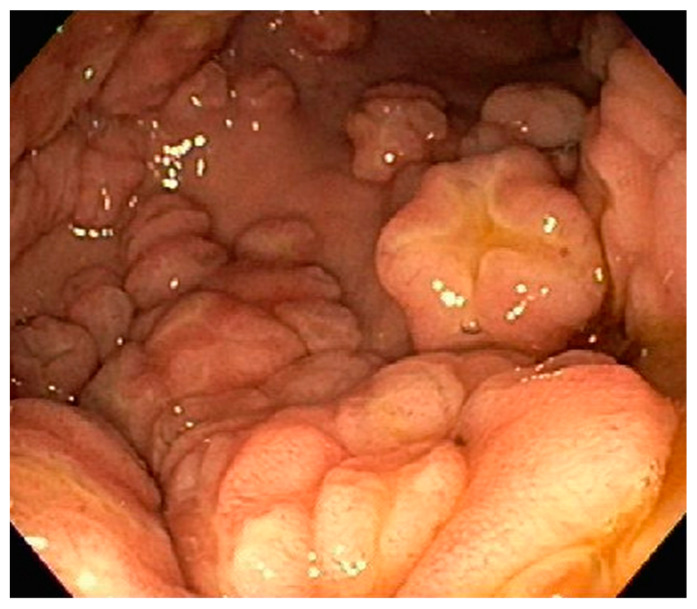
Innumerable polypoid appearing gastric lesions throughout the stomach.

**Figure 16 diseases-10-00061-f016:**
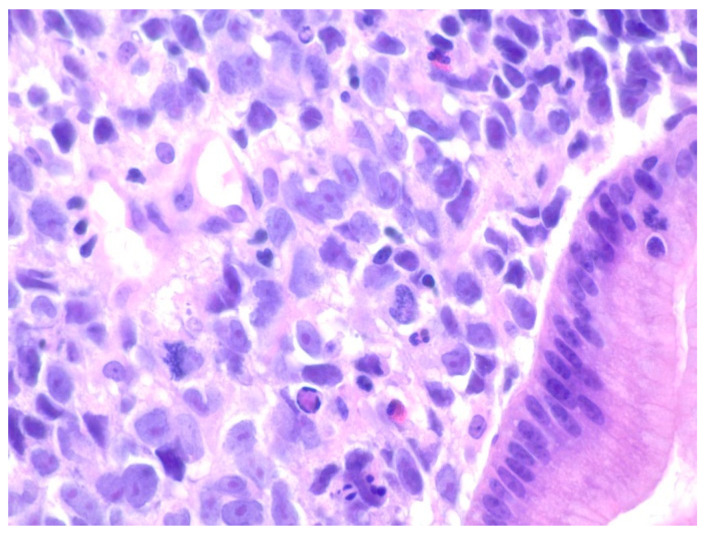
Hematoxylin and eosin-stained section showing pleomorphic cells with prominent nucleoli with intranuclear cytoplasmic pseudoinclusion and atypical mitosis suggestive of malignant melanoma in a background of gastric foveolar epithelium at 600× magnification.

**Figure 17 diseases-10-00061-f017:**
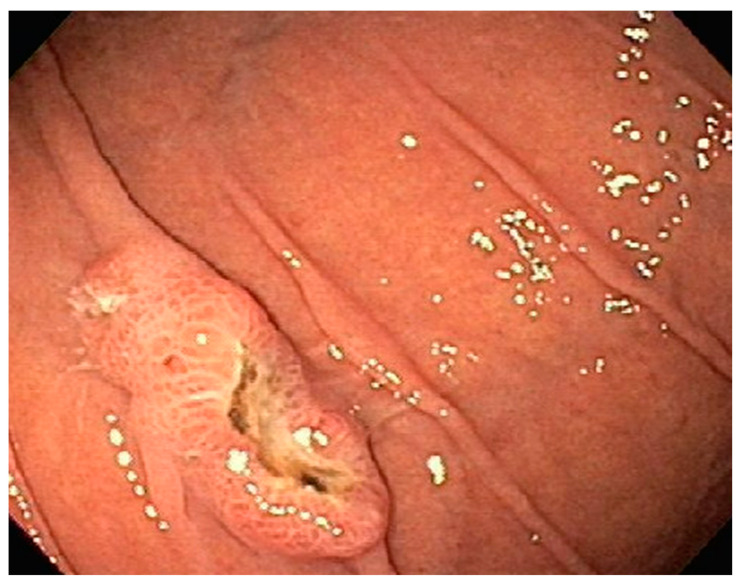
Large cratered ulcer seen in the gastric body.

**Figure 18 diseases-10-00061-f018:**
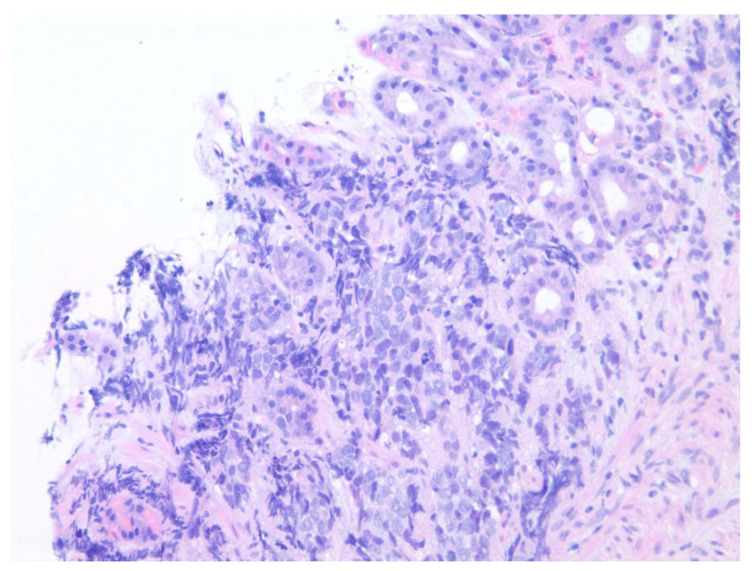
Poorly differentiated carcinoma with nested growth of small cells, with scant cytoplasm, small, oval to elongated nuclei, evenly distributed smudged chromatin, inconspicuous nucleoli, nuclear molding and brisk mitotic activity consistent with small cell carcinoma seen on on Hematoxylin and eosin stain at 200× magnification.

**Table 1 diseases-10-00061-t001:** Patient Characteristics.

	Sex	Age	Primary Cancer	Tumor Location	Gross Appearance	Additional Metastases	Pathology	Treatment	Outcome
1	F	66	Skin	Fundus, cardia, and body	Polypoid lesions with superficial ulcers	Esophagus, duodenum, mediastinum, cervical soft tissue, pancreas	Metastatic malignant melanoma	Radiotherapy and immunotherapy	Comfort care and death 5 days after EGD diagnosis of metastasis
2	M	84	Lymphoid	Fundus and cardia	Ulcerated and necrotic mass	Liver, urinary bladder	Diffuse large B-cell lymphoma	None	Death in 2 months after EGD diagnosis of metastasis
3	F	35	Lung	Fundus	Fungating mass	Esophagus, mediastinum	Large-cell neuroendocrine carcinoma	Chemotherapy-carboplatin/ etoposide	Death 1 month after EGD diagnosis of metastasis
4	F	73	Kidney	Fundus and body	Multiple large friable masses	Liver and lungs	Clear cell renal carcinoma	Nephrectomy, chemotherapy, immunotherapy (Axitinib)	Death 10 days after EGD diagnosis of metastasis
5	M	76	Lung	Body	Cratered ulcers	Liver, bone and adrenals	Small cell carcinoma (poorly differentiated neuroendocrine Ca)	None	Death 7 days after EGD diagnosis of metastasis
6	M	61	Lymphoid	Unknown	Clean based ulcer	None	Marginal zone lymphoma	Rituximab, Zanubrutinib, umbrasilib. OCHOP regimen currently	Progressive disease per latest scans despite therapy

## Data Availability

The data presented in this study are available on request from the corresponding author.

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
