# Peer review of "A Case Series of Gastric Metastatic Growths"

_diseases, 2022, doi:10.3390/diseases10030061_

Round 1
Reviewer 1 Report
Position of references should be before the stop sign (.). Fix the reference positions in line 22, 27, and 194.
Line 24-27: The sentence is confusing. Last part of this sentence is not connected to the previous statement within the same sentence. The authors should split the sentence into two sentences to make it more meaningful.
Line 48: correct the statement…Table I to Table 1
Line 65: Did the author organize Table 1 based on certain characteristics? If not, just organize the table based on gender or primary cancer location.
Please include the scale size in the figures.
Author Response
Response to Reviewer # 1:
Firstly, we would like to thank the reviewer for taking the time to review and provide suggestions for improvement.
1) Position of references should be before the stop sign (.). Fix the reference positions in line 22, 27, and 194.
Completed.
2) Line 24-27: The sentence is confusing. Last part of this sentence is not connected to the previous statement within the same sentence. The authors should split the sentence into two sentences to make it more meaningful.
Completed.
3) Line 48: correct the statement…Table I to Table 1
Corrected.
4) Line 65: Did the author organize Table 1 based on certain characteristics? If not, just organize the table based on gender or primary cancer location.
It was to outline all the individual cases for which we had the details.
5) Please include the scale size in the figures.
Unfortunately, we do not have the scale available for the images.

Reviewer 2 Report
This is a well-presented, in many aspects, case-series manuscript, and as such, this should be reflected in the title. ? in titles should overall be avoided, especially when the stem is not associated with a brief response. Furthermore, some readers may find it repulsive to have a question only with which sometimes don't agree. So, instead of 'Glorifying gastric growths', I would recommend traditionally naming your manuscript " A case series of....".
As for the presentation, good; I only found the images thrown in disorienting (too big and better to be presented in a panel and per category, i.e., endoscopic/histology etc). please.
Author Response
Response to Reviewer # 2:
Firstly, we would like to thank the reviewer for taking the time to review and provide suggestions for improvement.
1) This is a well-presented, in many aspects, case-series manuscript, and as such, this should be reflected in the title. ? in titles should overall be avoided, especially when the stem is not associated with a brief response. Furthermore, some readers may find it repulsive to have a question only with which sometimes don't agree. So, instead of 'Glorifying gastric growths', I would recommend traditionally naming your manuscript " A case series of....".
Changed.
2) As for the presentation, good; I only found the images thrown in disorienting (too big and better to be presented in a panel and per category, i.e., endoscopic/histology etc). please.
Have adjusted the size of the images. The images are presented after details and description of every case and would be very challenging to create such a table and would be large and difficult to read.

Reviewer 3 Report
Summary:
Gastric metastases are rare events, occur at a late stage of the neoplastic disease, and have a poor prognosis. Diagnosis of gastric metastases is problematic because they simulate primary gastric malignancy at both endoscopic and microscopic levels. A correct diagnosis is based on good communication between gastroenterologists and pathologists.
This study aimed to demonstrate the prognostic impact of a comprehensive evaluation and appropriate treatment for gastric metastatic patients’ outcome.
The authors examined the clinicopathological features of metastatic tumors in the stomach from distant sites in a small series of cases adding further clarification of the still largely elusive picture of the gastric metastasis.
The genesis of the article is appropriate to understand the extent of gastric metastases followed by malignancy of distant organs. However, to be accepted for publication some issues must be clarified and revised.
Peer-review Report:
In a general way, the paper is written in a simple way addressing a concept that still needs further investigation.
Despite this starting point, some topics must be addressed and defined in a more detailed manner in order to accomplish the standards of the journal.
Minor issues:
1. The title doesn’t sound appropriate for the content of the paper. I suggest adjusting the title so that the reader is referred to gastric metastases as a late consequence of other primary malignant events.
2. The abstract is concise; however, it does not accurately summarize the essential information of the paper. I strongly recommend that the aim, results and conclusions of the present study are included in a clearer and succinct way.
3. The introduction nicely summarizes present concerns regarding cancer metastases to the stomach. Still, the bibliography should be up to date in order to include the most recent Global Cancer Observatory estimations for GC (2020). More specifically, in line 22, in addition to reference [1], the aforementioned reference must be added.
4. Through the introduction, the authors placed the period not at the end of the sentence, but before the bibliographic references. This gap must be amended in the lines 22 and 27.
5. The sentence that is presented by the authors in lines 32 and 33 has a minor recurrence of vocabulary (present/presentations) which makes reading monotonous. The same happens with the sentence in lines 88 to 90 where the word “later” appears repeatedly. I suggest that authors come up with alternative terms avoiding repetition.
6. Throughout the text there are many flaws with regard to abbreviations. Either the abbreviations are not mentioned the first time the word is written, or they appear as abbreviated when their description has not been previously mentioned. To ensure the text is cohesive, authors should correct this lapse considering the words such as “magnetic resonance imaging” and “positron emission tomography”, “overall survival” and abbreviations such as “FDG”, “TTF-1”, “VATS”, “RCC”, “GE” and “NHL”.
7. To improve the description of the methods the article would benefit from the inclusion of an extra section where the authors could describe in detail the approaches used for staining tissue sections as well as the antibodies involved in the immunohistochemistry.
8. For a better reading fluidity and a clear interpretation of the results, whenever the authors refer to a patient, I recommend that they identify the patient's number as shown in table 1 throughout the text (including results and discussion sections). Moreover, the detailed description of patient’s characteristics should not be presented randomly. I suggest that the authors describe the patients taking into account some sort of order, such as age, gender, disease severity, outcome, etc.
9. The article would also be enhanced if the authors were able to present the images in the form of a schema for each of the patients. That is, instead of presenting each image as an individual figure, produce a set of images per patient. In addition to making the article much more appealing, it will help the reader to have a better understanding of the results.
10. It should also be noted that the images have low resolution, specially the ECG and endoscopic images. If possible, I advise that the images can be presented with a better definition.
To follow this issue, the hematoxylin and eosin-stained and immunohistochemistry images are too large not being presented with the same magnification. So that the article is presented in a more consistent way, I suggest that these images have the same magnification, and if there is any detail that the authors consider relevant, add a zoom in box highlighting that detail.
11. The sentence written between lines 70 and 74 creates some confusion for the reader. As an alternative, I suggested that they be reformulated as just two sentences that would end one in Figure 2 and the other in Figure 3.
12. The legend of Figure 10 is not very clear, I recommend that you change the description of it. Concurrently, the words showing and atypical appear together with no spacing between them. This spelling error should also be fixed.
13. In the description of the characteristics of patient 1 (66-year-old female), the paper would benefit from the inclusion of the morphological alterations of the gastric biopsies and that these are not only described in the legend of Figure 16.
14. In line 155 and 156, the authors must delete one of the repeated words “Imaging had revealed” or “Imaging revealed”.
15. To ameliorate the discussion of the paper, the authors should identify reports that corroborate melanoma as one of the most common malignancies to metastasize to the gastrointestinal tract.
16. The last paragraph of the discussion of the article should be improved so that the results obtained are enhanced. Authors should be able to clearly and concisely show the importance of detecting gastric metastases as a consequence of other primary cancers, the prognosis of these patients and the therapeutic approaches given that they are detected at a late stage of the disease.
Author Response
Response to Reviewer # 3:
Firstly, we would like to thank the reviewer for taking the time to review and provide suggestions for improvement.
Minor issues:
- The title doesn’t sound appropriate for the content of the paper. I suggest adjusting the title so that the reader is referred to gastric metastases as a late consequence of other primary malignant events.
Title has been changed.
- The abstract is concise; however, it does not accurately summarize the essential information of the paper. I strongly recommend that the aim, results and conclusions of the present study are included in a clearer and succinct way.
It has been modified.
- The introduction nicely summarizes present concerns regarding cancer metastases to the stomach. Still, the bibliography should be up to date in order to include the most recent Global Cancer Observatory estimations for GC (2020). More specifically, in line 22, in addition to reference [1], the aforementioned reference must be added.
Added reference.
- Through the introduction, the authors placed the period not at the end of the sentence, but before the bibliographic references. This gap must be amended in the lines 22 and 27.
Changed.
- The sentence that is presented by the authors in lines 32 and 33 has a minor recurrence of vocabulary (present/presentations) which makes reading monotonous. The same happens with the sentence in lines 88 to 90 where the word “later” appears repeatedly. I suggest that authors come up with alternative terms avoiding repetition.
Words have been changed.
- Throughout the text there are many flaws with regard to abbreviations. Either the abbreviations are not mentioned the first time the word is written, or they appear as abbreviated when their description has not been previously mentioned. To ensure the text is cohesive, authors should correct this lapse considering the words such as “magnetic resonance imaging” and “positron emission tomography”, “overall survival” and abbreviations such as “FDG”, “TTF-1”, “VATS”, “RCC”, “GE” and “NHL”.
Included under abbreviations section.
- To improve the description of the methods the article would benefit from the inclusion of an extra section where the authors could describe in detail the approaches used for staining tissue sections as well as the antibodies involved in the immunohistochemistry.
Unfortunately, that is beyond the intended scope of this article as it is to present the overall clinical aspect and would be difficult to obtain from our pathology colleagues in a short time.
- For a better reading fluidity and a clear interpretation of the results, whenever the authors refer to a patient, I recommend that they identify the patient's number as shown in table 1 throughout the text (including results and discussion sections). Moreover, the detailed description of patient’s characteristics should not be presented randomly. I suggest that the authors describe the patients taking into account some sort of order, such as age, gender, disease severity, outcome, etc.
We attempted to provide a brief overview of each patient case.
- The article would also be enhanced if the authors were able to present the images in the form of a schema for each of the patients. That is, instead of presenting each image as an individual figure, produce a set of images per patient. In addition to making the article much more appealing, it will help the reader to have a better understanding of the results.
We tried to combine them into panels but it does not look very appealing and came across as crowded.
- It should also be noted that the images have low resolution, specially the ECG and endoscopic images. If possible, I advise that the images can be presented with a better definition.
To follow this issue, the hematoxylin and eosin-stained and immunohistochemistry images are too large not being presented with the same magnification. So that the article is presented in a more consistent way, I suggest that these images have the same magnification, and if there is any detail that the authors consider relevant, add a zoom in box highlighting that detail.
Unfortunately, these are the best quality images we have overall from the sets and difficult to set the same scale for each of those images as the ideal ones were provided from both the endoscopy and histopathology standpoint.
- The sentence written between lines 70 and 74 creates some confusion for the reader. As an alternative, I suggested that they be reformulated as just two sentences that would end one in Figure 2 and the other in Figure 3.
Sentences have been modified.
- The legend of Figure 10 is not very clear, I recommend that you change the description of it. Concurrently, the words showing and atypical appear together with no spacing between them. This spelling error should also be fixed.
Figure legend has been modified.
- In the description of the characteristics of patient 1 (66-year-old female), the paper would benefit from the inclusion of the morphological alterations of the gastric biopsies and that these are not only described in the legend of Figure 16.
We apologize but are requesting clarification on this question.
- In line 155 and 156, the authors must delete one of the repeated words “Imaging had revealed” or “Imaging revealed”.
Correction has been made.
- To ameliorate the discussion of the paper, the authors should identify reports that corroborate melanoma as one of the most common malignancies to metastasize to the gastrointestinal tract.
We did include a few papers in this mini case series with references 6-9 involving melanoma.
- The last paragraph of the discussion of the article should be improved so that the results obtained are enhanced. Authors should be able to clearly and concisely show the importance of detecting gastric metastases as a consequence of other primary cancers, the prognosis of these patients and the therapeutic approaches given that they are detected at a late stage of the disease.
The paragraph has been modified.
